# Acute Effect of Riceberry Waffle Intake on Postprandial Glycemic Response in Healthy Subjects

**DOI:** 10.3390/foods10122937

**Published:** 2021-11-29

**Authors:** Patthamawadee Tongkaew, Deeyana Purong, Suraida Ngoh, Benjapor Phongnarisorn, Ebru Aydin

**Affiliations:** 1Department of Food Science and Nutrition, Faculty of Science and Technology, Prince of Songkla University Pattani Campus, Pattani 94000, Thailand; diyanapurong@gmail.com (D.P.); da.suraida08@gmail.com (S.N.); 2Department of Food Technology, Faculty of Agricultural Technology, Phuket Rajabhat University, Phuket 83000, Thailand; benjapor.p@pkru.ac.th; 3Department of Food Engineering, Suleyman Demirel University, Isparta 32260, Turkey; ebruaydin@sdu.edu.tr

**Keywords:** gluten-free product, Riceberry rice waffle, sucrose substitute, postprandial glycaemia

## Abstract

Gluten-free products have been developed due to increasing consumer demand. The improvement of the sensory quality and nutritional value of these products may support functional food development and provide health benefits. The purpose of this study was to develop a gluten-free waffle formulation with Riceberry rice flour by replacing the sucrose with maltitol and palm sugar powder. Evaluations of the sensory acceptability of these products and the blood glucose levels of healthy volunteers after consuming Riceberry and wheat flour waffles were carried out. The glycemic responses of the volunteers to the Riceberry and wheat flour waffles at 0, 15, 30, 45, 60, 90, 120, 150, and 180 min were monitored. In addition, the glycemic index of the products was calculated. The finding revealed that replacing sugar with 50% (*w/w* of total sugar) palm sugar powder and 50% maltitol was the most acceptable formulation that received the highest acceptability scores in terms of overall acceptability and texture. The blood glucose levels of both Riceberry waffle and wheat flour were not significantly different. The glycemic index of Riceberry waffle and wheat flour waffle were 94.73 ± 7.60 and 91.96 ± 6.93, respectively. Therefore, Riceberry waffle could be used as an alternative gluten-free product for celiac patients, but not for diabetic patients.

## 1. Introduction

Celiac disease (CD) is known to be a cause of severe malnutrition because it causes the inflammation of the small intestine after gluten consumption in some people. The prevalence of celiac disease has been reported to be approximately 1% worldwide and is expected to become higher [1,2]. Additionally, the prevalence of CD with T1DM has increased from 2.4% to 16.4% [3]. The only treatment available for celiac patients is to avoid foods containing gluten [4]. In general, gluten is a protein found in wheat, rye, and barley; hence, it affects the elastic properties of bread dough and bakery products [5]. Several approaches, such as approaches utilizing alternative flours, have been used to develop gluten-free products [5,6]. A mixture of gluten-free rice flour has been used in baked products to mimic the properties of wheat flour. However, products still often have unsatisfactory sensory qualities and poor nutritional value [6,7].

Waffles are a type of wheat-based baked product and are popularly consumed as breakfast or dessert items across the world. Waffles are usually made using a moderate to high content of table sugar [8]. Sugar not only gives waffles a sweet taste but also affects the viscosity and final texture of waffles [9]. A high consumption of sugar-rich foods has negative effects on health [10]. A reduction in consumption of sugar-rich foods may reduce one’s energy intake as well as one’s risk of overweight and obesity and metabolic syndromes such as diabetes [11]. To reduce the amount of sugar and calories contained in bakery products, sugar alcohol such as maltitol has been used as a sucrose substitute. Maltitol provides 90% of the sweetness but only half the calories of sucrose and has a very slow digestion rate [12]. Additionally, the substitution of sucrose for maltitol in sponge cake was found to not significantly affect its textural quality or acceptability [13]. Therefore, maltitol may be the most suitable sucrose replacement in bakery products [11]. Palm sugar is another alternative sweetener that is commonly used in Thai desserts due to its unique flavour and aroma, with a low glycemic index of 35. Furthermore, palm sugar exhibits better nutritional qualities and antioxidant properties than refined cane sugars [14,15,16].

After the consumption of carbohydrates, blood glucose levels increase based on the glycemic index (GI) of the food. GI is defined as an incremental area under the curve (iAUC) of the blood glucose after the consumption of food that contains 50 g of carbohydrates and is expressed as a percentage of the iAUC of 50 g of glucose in the same participant [10]. GI can be classified into three categories: low (<55), medium (55–69), and high (>70). The consumption of low-GI products is one of the major goals in preventing or treating diabetes mellitus. In addition, controlling blood glucose level by consumption of low GI food is preferable than lowering blood glucose by drug due to undesirable side effects. Recent studies have discovered that gluten intake affects both the microbiota and intestinal permeability, as gluten peptides cause an increase in inflammatory milieu after they are transported from the intestine. Based on the literature, following a gluten-free diet may play a role in reducing the risk of type 1 diabetes, whereas its effect on type 2 diabetes is less clear, with human intervention trials being needed to determine it [17].

Riceberry rice (*Oryza sativa* L.) is a type of dark purple rice that was created through crossbreeding between Khao Hom Nin Rice and Khao Hom Mali 105. The GI of Riceberry rice is 62 and it has recently been used as a functional ingredient in several food products, such as noodles, bread, pudding, and yogurt [18,19,20,21,22]. Gluten-free flour from this rice not only contains numerous nutrients but also phenolic compounds [20,22,23,24]. Additionally, the digestibility of Riceberry was tested in vitro and it was found to slow down glucose absorption after starch digestion [25]; however, there is a scarcity of information on the digestibility of Riceberry products in humans. For maltitol, the blood glucose and insulin response are low (35 and 27) [26], while the GI value of palm sugar is 35–42, which is considered to be a low value for GI and lower than the GI value of sucrose [15].

Therefore, the present study aimed to improve the physicochemical and sensory properties of gluten-free waffles by using Riceberry rice flour, maltitol, and palm sugar powder in the production of gluten-free waffles. Maltitol and palm sugar powder were used as alternative sweeteners instead of sucrose. The sensory quality and nutritional value of the waffles were considered. Furthermore, the acute effect of Riceberry (RB) waffle on postprandial glycemia was investigated and compared to that of a waffle made from wheat flour (WF), followed by the GI calculation of RB and WF waffles. The developed non-sucrose gluten-free and low-GI waffle could have reduced post-prandial glycemic effects when consumed; thus, they are suitable for celiac and diabetic patients as well as other health-conscious consumers.

## 2. Materials and Methods

### 2.1. Waffle Preparation

Riceberry grain, rice flour, glutinous rice flour, tapioca flour, caster sugar, palm sugar powder, and salt were purchased from a local supermarket (Supercheap, Phuket, Thailand). Baking powder, xanthan gum, and maltitol were purchased from the Gerbera house shop (Phuket, Thailand). Riceberry flour was made by grinding grain for 5 min at setting 5 (EM-11, Sharp, Shah Alam, Malaysia). The dry ingredients were weighted according to the formulation shown in Table 1 and mixed by blending for 3 min. The dry ingredients of each formulation were mixed with 60 g of water, 50 g of egg, and 10 g of vegetable oil to make a batter. The batter was added to a waffle maker (HOM-WS06, Versu, Bangkok, Thailand) and cooked for 4 min at 180–190 °C.

### 2.2. Sensory Evaluation

Nine-point hedonic sensory evaluations were performed with 50 untrained volunteer panels; volunteers were both male and female and were recruited from the university staff and undergraduate students. Briefly, the waffles were cooked in the manner described in Section 2.1: “Waffle Preparation”. Next, approximately 5 g of waffles of each formulation were assigned a 3-code digit and randomly provided to the panels. A cup of water was provided to the volunteers so that they could rinse their mouths between tasting each sample. The acceptability of the waffle was evaluated in terms of color, texture, aroma, taste, and overall acceptability. Subsequently, the most acceptable formulation was selected for use in a human study.

### 2.3. Participants

The digestibility of the acceptable Riceberry waffle and wheat flour waffle (Table 2) was tested in humans in accordance with the ethical principles of the Declaration of Helsinki and Belmont Report and approved by the Human Research Ethics Committee of Prince of Songkla University, Pattani Campus (REC Number: psu.pn. 1-001/62). Participants were recruited from Prince of Songkla University, Pattani Campus through posters and social media advertisements (Facebook, Line group). The inclusion criteria were considered as follows: age of 35–55 years, body mass index of 18.5–24.9 kg/m^2^, fasting blood sugar level of <100 mg/dL, and normal waist circumference ≤90 cm for males and ≤80 cm for females. Subjects with chronic diseases (e.g., diabetes, blood pressure, and chronic kidney dis-eases), gastrointestinal problems, currently took medications, had allergies or sensitivity to the product’s ingredients, had donated blood in the prior 2 months, pregnant and/or breastfeeding were excluded from this study. The sample size of 10 subjects was determined according to Brouns et al. [27]. The purpose, content, and safety of the study were fully explained to all subjects. Before the human intervention study began, all participants provided their written informed consent. The volunteers were given an incentive after they completed the study. The baseline characteristics of the 10 subjects are presented in Table 3.

### 2.4. Study Design

This study was a single-blind controlled study. The study consisted of three experimental trials (two test meals and one reference) carried out on three different occasions in accordance with the literature [28,29,30,31,32]. We allowed a one-week wash out period between each visit. The first test meal consisted of 111 g of Riceberry waffle consumed with 250 mL of drinking water. The second test meal consisted of 111 g of wheat flour waffle consumed with 250 mL of drinking water. The test meals contained 50 g of available carbohydrate, of which two pieces of approximately 11 cm × 10.5 cm × 0.9 cm (wide × long × thick) were provided (Figure 1). The reference meals consisted of 50 g of glucose mixed with 150 mL of warm water consumed with 250 mL of drinking water. All test meals were consumed within 15 min. The proximate analysis of both waffle samples was performed and the results are listed in Table 4.

The day before each visit, subjects were told to avoid consuming food with high levels of dietary fiber, food with a high polyphenol content (e.g., vegetables and fruits), alcohol, caffeinated beverages (e.g., coffee and sparkling water), smoking, and exercise. Additionally, subjects were asked to fast for 10–12 h. In the morning of each visit, subjects arrived at the study center and their capillary blood glucose was taken at the fasting state (0 min) before the consumption of food samples and 15, 30, 45, 60, 90, 120, 150, and 180 min after a test meal. Capillary blood glucose was acquired through the puncture of the ventral finger skin and measured with the ACCU-CHEK^®^ Performa (Roche Diabetes Care, Mannheim, Germany). Blood sugar levels after the consumption of the glucose solution was used to calculate the iAUC and GI of waffles. The study design is described in Figure 2.

### 2.5. Statistical Analysis

The results are presented as means ± standard deviations. The differences between averages were determined through analysis of variance (ANOVA) tests followed by Duncan’s Multiple Range Test (DMRT) at a 95% confidence interval. A Randomized Complete Block Design (RCBD) was used for the sensory analysis. 

The iAUC was calculated based on the trapezoidal rule, where the area of all glucose responses that were collected for three hours were added together after the area beneath the baseline was ignored using Microsoft Excel 2013. Average comparisons of blood sugar levels before and after the consumption of Riceberry waffle, wheat flour waffle, and glucose at different time points were made using the analysis of variance (ANOVA) test. The glycemic index (GI) was calculated by Alongi et al. [35]. The average comparison of the glycemic index of Riceberry waffle and wheat flour waffle was analyzed with a *t*-test. All statistical analyses were completed with SPSS version 25 (IBM, Armonk, NY, USA). Statistical significance was achieved at a 95% confidence interval (*p* ≤ 0.05).

## 3. Results

Cereal grains are generally used as major raw materials in waffles, as they impart an elastic or chewy texture [8]. Therefore, in order to obtain the desired waffle characteristics, the RB flour was developed as a ratio of tapioca and glutinous rice flour. Additionally, the sucrose content of the waffle was substituted by different sugars to achieve a lower postprandial blood glucose level.

### 3.1. Sensory Evaluation

Riceberry flour was used as an alternative ingredient in waffle formulation and adapted by replacing the sucrose with maltitol and palm sugar powder. Four formulations with 100% sucrose (as control), 100% maltitol, 50% sucrose to 50% maltitol, and 50% palm sugar powder to 50% maltitol were prepared according to Table 1. The sensory aspects of the gluten-free flour formulation were evaluated and are shown in Table 5.

There was no significant difference between each treatment in terms of appearance and color, indicating that maltitol and palm sugar did not decrease the acceptability of the waffle in terms of appearance and color. However, in terms of aroma and taste, 100% maltitol contributed to a significant decrease compared with the control and other recipes. The texture score of the 100% maltitol waffle was lower than that of the waffle with 50% palm sugar powder and 50% maltitol (*p* ≤ 0.05). The waffle with 50% palm sugar powder and 50% maltitol was the most acceptable and received the highest acceptability score in terms of overall acceptability and texture. Moreover, the score of all attributes of this formulation was not significantly different from that of the Riceberry waffle made with 100% sucrose (control). On the other hand, the aroma, taste, and overall acceptability scores of waffles made with 100% maltitol were significantly lowered than those of the control samples (*p* ≤ 0.05).

Hence, the 50% palm sugar powder and 50% maltitol was found to be a suitable substitute for sucrose in waffles made with Riceberry flour and this recipe was selected for use in an intervention study aiming to observe the glycemic response of gluten-free Riceberry waffle.

### 3.2. Postprandial Blood Glucose and Glycemic Index

The Riceberry (dark brown) and wheat flour waffles (white) were prepared and are shown in Figure 1. There was not a significant difference between each type of waffle in terms of nutritional composition, except for the ash and moisture content. The content of carbohydrate, crude fat, protein, etc., in 100 g of waffle are presented in Table 4.

Descriptive characteristics of subjects are summarized in Table 3. It was found that both the fasting glucose concentrations and anthropometric data of participants were within normal ranges. The blood glucose levels of participants after the consumption of Riceberry waffle, wheat flour waffle, and glucose solution at 15, 30, 45, 60, 120, 150, and 180 min are shown in Figure 3.

The blood levels of participants after the consumption of glucose solution and Riceberry waffle peaked at 30 min. However, the highest level was observed at 45 min for the waffle made from wheat flour. Subsequently, the blood glucose levels of the participants slowly decreased until 180 min in all treatments (Figure 3A). No significant difference in blood glucose levels was found after the consumption of the Riceberry waffle compared to after the consumption of the wheat flour waffle (Figure 3B).

The incremental area under the curve (iAUC) from 0 to 180 min was 9824.92 ± 946.82, 9536.92 ± 678.63, and 10,220.92 ± 744.72 mg/dL min for the RB waffle, WF waffle, and glucose solution, respectively, as represented in Figure 3B. The glycemic index of both samples was calculated. The glycemic index of Riceberry waffle and wheat flour waffle was 94.73 ± 7.60 and 91.96 ± 6.93, respectively, indicating a high glycemic level.

## 4. Discussion

### 4.1. Sensory Aspect

The sensory evaluations were performed based on the appearance, color, aroma, taste, and texture acceptability of the RB and WF waffles. The given highest score of each measuring total indicated which waffle had the highest acceptability. Based on sensory analysis, the acceptability of the waffles in terms of appearance and color did not differ between the formulations. This is possibly due to the similarity of the melting point (145 °C) of maltitol and sucrose. Additionally, after exposure to high heat, maltitol liquifies, caramelizes, and turns brown in color similarly to sucrose [36]. In addition, it was discovered that the replacement of sucrose with alternative carbohydrates had no adverse effect on the overall appearance and color of the waffles. According to our sensory analysis, the 100% maltitol waffle samples differed significantly in terms of aroma and taste compared to the other samples (Table 5). This observation may be explained by the fact that sucrose is sweeter than maltitol [37]. Based on texture, the waffle samples differed significantly from the control sample. The highest scores for the texture were obtained when 50% palm sugar powder and 50% maltitol were used, while the 100% maltitol waffle sample had the lowest texture score. The increase in the amount of maltitol at a higher level of substitution possibly caused an increase in the hardness of the waffle. As shown in Table 4, the use of 100% maltitol led to a significant decrease in the texture acceptability score compared with the other formulations. The result found in this study is in line with that found by Sukhonthara et al. [25]. They found that increasing the amount of maltitol in toddy palm cakes caused an increase in the cakes’ firmness and toughness [25]. By utilizing 100% maltitol in sponge cake, not only were the sensory properties of the baked products affected, but the physical properties were also affected. The use of 100% maltitol resulted in a significant decrease in volume and increase in water activity, firmness, and rate of becoming stale compared to products made with sucrose [38]. On the other hand, the use of palm sugar was found to decrease the hardness and adhesiveness of wheat flour bread [39]. Therefore, the use of palm sugar powder may compensate for the hardness caused by maltitol. The water retention ability of fructose and glucose from palm sugar power may promote the retention of water and preserve the product’s moisture [14,26,40]. Hence, waffles made with this formulation contain more moisture, which is a softening agent, resulting in waffles made with palm sugar having a lower hardness.

In a literature review, the sensory quality of noodles made by substituting wheat flour with Riceberry flour was studied and it was found that noodles made with a 30% substitution were not significantly different from noodles made with 100% wheat flour in terms of flavor, taste, softness, stickiness, and overall acceptability [18]. In addition, it was shown that the use of Riceberry flour in noodles did not affect customer satisfaction [18]. In the current study, approximately 27% Riceberry flour was used in the waffle mixture and their acceptability score was “slightly like” to “moderately like”. Therefore, consumers are likely to buy instant mixes of Riceberry flour waffle due to its high iron level and low glucose content, making it suitable for anemic and diabetic patients [41]. In addition, the substitution of sucrose with maltitol and/or palm sugar powder may also improve the decrease seen in postprandial blood glucose levels.

### 4.2. Human Study Aspect

This is the first study to develop a gluten-free Riceberry waffle and investigate the glycemic response to its ingestion in humans. The dark brown appearance of the Riceberry waffles in this study was due to a polyphenol called anthocyanin, which are present in high amounts in Riceberry flour [17,18]. The color of food can influence consumers’ appetites. For instance, white and yellow soups attract consumers, and consumers’ willingness to consume blue soup was found to be significantly decreased compared their willingness to consume white and yellow soups [42]. However, the consumption of Riceberry waffles possibly provides health benefits due to the anthocyanin content of Riceberry flour [20,22,43,44]. In a preliminary study, we found that a Riceberry waffle instant mix (consisting of dry ingredients that can be made into a batter) contained total phenolic compounds of 3014 µg/g gallic acid equivalent, while cooked Riceberry waffles contained total phenolic compounds of 83 µg/g gallic acid equivalent. The amount of total phenolic compounds was reduced due to the heat used during cooking [45]. Additionally, the bioavailability and health benefits of anthocyanin in Riceberry waffles need to be further studied. A loss of anthocyanin content after cooking may reduce the biological activity of polyphenols in carbohydrate metabolism and fail to decrease the blood glucose level. In Table 3, it is evident that the Riceberry waffle contained more ash than the wheat flour waffle. Therefore, the consumption of Riceberry waffles is not only capable of providing benefits from anthocyanin but also benefits of other nutrients. For example, Riceberry contains zinc (3.2 mg/100 g), which can reduce glucose absorption while promoting glucose metabolism and storage [46]. Diabetic patients must monitor their postprandial blood glucose level and avoid consuming foods with a high glycemic index to prevent a rapid increase in their blood sugar level [47]. Additionally, Riceberry flour does not contain gluten and some studies have reported that gluten-free diets can help to prevent diabetes by reducing leptin and insulin resistance and increasing beta cell volume [17]. Since Riceberry flour is rich in anthocyanidins and zinc and does not contain gluten, it may be useful to think about its potential use in gluten-free waffles to prevent customers from experiencing hyperglycemic peaks.

In this regard, a human study was planned to explore the effect of the consumption of 100 g of RB waffle on postprandial glucose in this research. This human study revealed that wheat flour was absorbed more slowly into the blood than Riceberry flour or glucose. This may be due to the food matrix of the waffle, making it harder to digest and absorb than the free form of glucose [48]. Furthermore, this study revealed that glucose from the wheat flour waffle was absorbed more slowly (at 45 min) than glucose from the Riceberry waffle (at 30 min). Based on its digestibility, starch can be classified into three groups: resistant starch, rapidly digestible starch, and slowly digestible starch [1]. The digestibility of different types of starch is possibly caused by the rate and duration of the glycemic response [49]. To improve the structure and texture of gluten-free products, different sources of food starch have been combined into different food formulations [1]. In this study, Riceberry rice flour, two major types of food starch, and a minor type of starch—tapioca, corn, and glutinous rice flour—were combined to create the Riceberry instant mix formulation (Table 1). Based on our literature review, the combination of mixtures used in gluten-free products may cause an increase in postprandial blood glucose [43]. For example, adding corn or potato starch and rice flour to gluten-free bread provides high GI values ranging from 83.3 to 96.1 [43]. Therefore, the combination of flours used in the waffle formulation in this study probably caused rapid hydrolysis, resulting in a high GI value. However, some studies have shown that foods made with Riceberry rice have a low GI. For instance, the GI of Riceberry pudding was discovered to be 46 and, in another study, the GIs of bread made from Riceberry, wheat, and Hom Mali were 69.3 ± 4.4, 77.8 ± 4.6, and 130.6 ± 7.9, respectively [19,43,44]. The GI of food depends on various factors—for example, the type of starch and fiber used and how finely milled the flour is. Rice with a high amylose content has a lower glycemic index than low-amylose varieties [43,47]. The GI value of rice products is about 100, and this value is in accordance with the formation and rate of digestion of resistant starch (RS). After the consumption of food containing RS over 5–7 h, the postprandial glycaemia and insulinemia values trend to decrease [49,50,51,52]. Additionally, it has been reported that the use of high-amylose wheat flour in noodles reduces glycemic values in humans [53].

In this study, the iAUC and cMax are consistent with the blood glucose level of each treatment. The blood glucose level of free sugar was reduced to baseline within 2 h, while that of waffle samples was clear within 3 h. This may be because of their dietary fiber content. The digestibility of waffles may be increased by the use of different types of starch and may be affected by the duration of glycemic response [42]. Additionally, the glycemic indexes of Riceberry waffle and wheat flour are at high. The might be because of the flour mixture used in RB waffles (tapioca, corn, and glutinous rice flour). The carbohydrate content of different types of roasted tapioca flour was found to be between 95.9 and 89.9%, while the starch content was found to be between 86.9 and 94.75% [54]. The major carbohydrate in glutinous rice flour has been reported to be amylose, and it contains 88.8% starch and 0.84% free sugar [55]. The last flour used in the RB waffle recipe was corn flour, and its carbohydrate content is 62.38% [56]. Therefore, RB waffle has a high GI and, based on this, it may be concluded that the postprandial blood glucose level of volunteers was not decreased after the consumption of RB waffle compared to the consumption of glucose and WB waffle. To provide the ideal texture and acceptability of the waffle, the flour mixture needed to be used in the RB waffle formulation [9].

Although the glycemic index and glycemic load of Riceberry waffle are still high and it may not appropriate for controlling the blood glucose level, it is still a functional product used for celiac patients. In addition, compared to other gluten-free waffle products, as presented in Table 4, the RB waffle has a lower energy value and carbohydrate content. Therefore, the RB waffle may be a better product for celiac patients based on its energy intake and phenolic content values. It is also a good choice for use as a weight management product in overweight or obese patients due to its low energy values. Furthermore, this product might be useful for controlling other biomarkers, such as improving the plasma antioxidant capacity and decreasing lipid peroxidation [22]. Based on the literature review, numerous gluten-free products such as bread and cake have been developed [6]. However, information on gluten-free waffle products is scarce. This study was the first to develop gluten-free waffles. Additionally, the test of the digestibility of the Riceberry waffle was performed in humans. Ten volunteers are a sufficient number for studying glycemic index, but it would be better to measure the plasma insulin level of participants along with their blood glucose level in order to understand the association. High standard deviation at each plasma glucose measurement could be due to the mixed gender of participants (female participants tend to have higher plasma glucose responses than males [52,57]), the number of participants [27,58] and/or measurement errors with glucometer [58]. However, recently some clinical studies have been published in the literature that use only 10 participants for measuring postprandial glycemic response [59,60,61,62]. Additionally, due to its convenience in application for the participants, a glucometer was used in most of these studies [28,29,30,31,32]. In addition, it was reported that the use of a glucometer for monitoring the postprandial blood glucose level has an acceptable sensitivity [63]. The preliminary clinical results of this study suggest that the intake of RB waffles does not increase postprandial blood glucose levels. Nevertheless, it is reasonable to consider exploring Riceberry flour’s health-promoting activities due to its high phenolic content in the functional food industry.

## 5. Conclusions

The waffle made from Riceberry flour with the substitution of palm sugar and maltitol for sugar was found to be acceptable to consumers in terms of taste and texture. The waffle also contained a higher fiber and mineral content than waffles made with wheat flour. The postprandial glycemic responses of participants to the consumption of the Riceberry flour waffle and the wheat flour waffle did not significantly differ. The developed product may be a gluten-free alternative for CD patients but it still not suitable for consumption by diabetic patients. Overall, the present research indicates that gluten-free flour made from Riceberry may be an alternative valuable source for use in functional food products and could be recommended for celiac disease patients. It is also important to further explore the role of gluten-free diets and the use of antioxidant-rich gluten-free flours in diabetic people.

## Figures and Tables

**Figure 1 foods-10-02937-f001:**
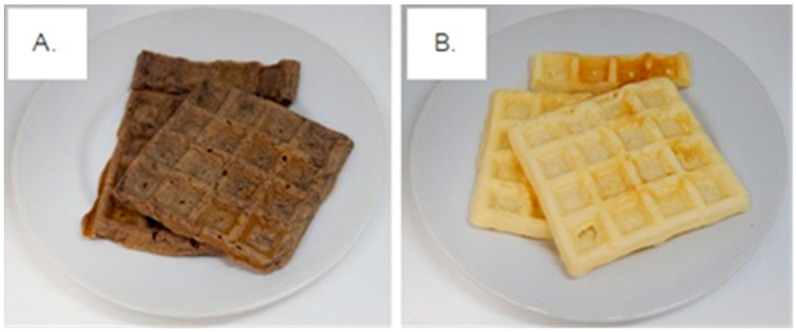
(**A**) Riceberry waffle (RB) and (**B**) wheat waffle (WF) provided to the volunteers.

**Figure 2 foods-10-02937-f002:**
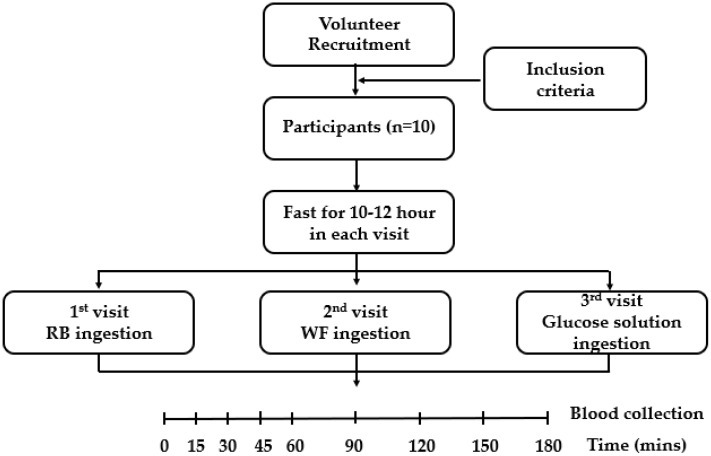
The study design.

**Figure 3 foods-10-02937-f003:**
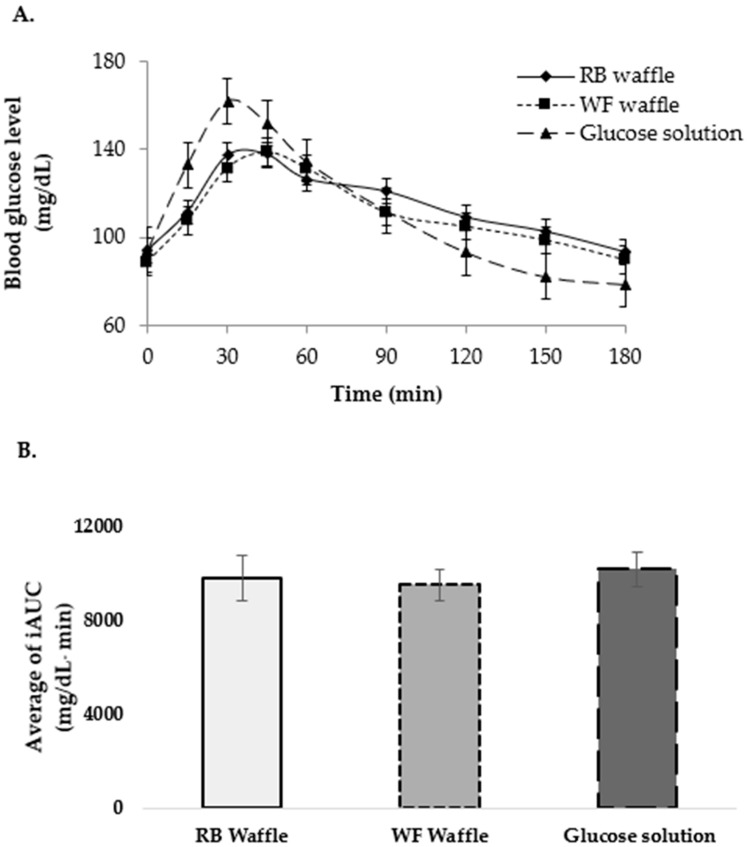
(**A**) The blood glucose level (**B**) the incremental area under the curve from 0 to 180 min (iAUC 0–180 min) after the ingestion of Riceberry waffle, wheat flour waffle, and glucose solution (*n* = 10).

**Table 1 foods-10-02937-t001:** Waffle formula replaced sucrose with maltitol and palm sugar powder at different percentages.

Ingredients (g)	100% Sucrose (Control)	100% Maltitol	50% Sucrose + 50% Maltitol	50% Palm Sugar Powder + 50% Maltitol
Riceberry flour	30	30	30	30
Tapioca flour	15	15	15	15
Corn flour	10	10	10	10
Glutinous rice flour	15	15	15	15
Baking powder	3	3	3	3
Salt	1.5	1.5	1.5	1.5
Cane Sugar (sucrose)	22	-	11	-
Milk powder	15	15	15	15
Xanthan gum	0.5	0.5	0.5	0.5
Palm sugar powder	0	0	0	11
Maltitol	0	22	11	11

**Table 2 foods-10-02937-t002:** Riceberry (RB) and wheat flour (WF) waffle formula used for the digestibility study in humans.

Ingredients (g)	RB Waffle	WF Waffle
Wheat flour	-	60.5
Riceberry flour	30	-
Tapioca flour	15	-
Corn flour	10	10
Glutinous rice flour	15	-
Baking powder	3	3
Salt	1.5	1.5
Cane Sugar (sucrose)	-	22
Milk powder	15	15
Xanthan gum	0.5	-
Palm sugar powder	11	-
Maltitol	11	-

**Table 3 foods-10-02937-t003:** Subject characteristics at baseline.

Characteristic	Males (*n* = 5)	Females (*n* = 5)
Age (years)	22.2 ± 4.34	22.2 ± 0.45
Weight (kg)	63.2 ± 3.42	50.0 ± 4.69
Height (cm)	173.2 ± 3.42	156.2 ± 5.49
Body mass index (BMI, kg/m^2^)	21.09 ± 1.71	20.32 ± 1.41
Fasting blood sugar (FBS, mg/dL)	91.8 ± 8.61	95.8 ± 2.71
Waist circumference (cm)	78.8 ± 5.22	70.4 ± 4.50

Values are shown as mean ± S.D.

**Table 4 foods-10-02937-t004:** Nutrition composition of the Riceberry (RB) and wheat flour (WF) waffles per 100 g.

Nutrition Composition	Proximate Analysis		Nutrition Database	Nutrition Label
	RB Waffle	WF Waffle	Gluten-Free Flour (Missbach et al. [33])	Gluten-Free Flour (Hughes et al. [34])
Energy * (kcal)	275.82	268.86	345.50	353.11
Protein (g)	6.54	6.04	3.70	6.60
Fat ** (g)	7.70	7.10	1.90	1.80
Carbohydrate (g)	45.09	45.20	77.60	76.90
Sugars (g)	-	-	-	0.80
Fiber *** (g)	0.12	0.09	3.80	3.00
Sodium (mg)	-	-	-	5.00
Ash (g)	2.30	1.02	-	-
Moisture (g)	38.37	40.64	-	-

* The energy content of waffles was calculated from macronutrients (protein = 4 kcal/g, fat = 9 kcal/g, and carbohydrate = 4 kcal/g). ** In this study, the values represent crude fat, while other references refer to total fat plus saturated fat. *** In this study, the values represent crude fiber, while other references refer to dietary fiber.

**Table 5 foods-10-02937-t005:** Sensory evaluation scores for Riceberry waffle made with different sweeteners.

Attributes	100% Sucrose (Control)	100% Maltitol	50% Sucrose + 50% Maltitol	50% Palm Sugar Powder + 50% Maltitol	F-Value
Appearance	6.84 ± 1.22 ^a^	6.84 ± 1.25 ^a^	6.88 ± 1.51 ^a^	6.76 ± 1.35 ^a^	0.19
Colour	6.86 ± 1.15 ^a^	6.75 ± 1.28 ^a^	6.65 ± 1.37 ^a^	6.67 ± 1.37 ^a^	0.83
Aroma	6.45 ± 1.25 ^a^	5.94 ± 1.69 ^b^	6.53 ± 1.29 ^a^	6.59 ± 1.46 ^a^	6.56
Taste	6.49 ± 1.65 ^a^	5.71 ± 1.89 ^b^	6.79 ± 1.27 ^a^	6.59 ± 1.69 ^a^	10.64
Texture	6.57 ± 1.62 ^ab^	6.31 ± 1.53 ^b^	6.78 ± 1.25 ^a^	6.94 ± 1.45 ^a^	3.97
Overall acceptability	6.82 ± 1.42 ^a^	6.24 ± 1.69 ^b^	6.94 ± 1.29 ^a^	6.98 ± 1.41 ^a^	8.36

All the values are means ± standard deviations (SDs) of 50 participants. In each attribute row, values with different letters are significantly different; *p* ≤ 0.05.

## Data Availability

Data sharing not applicable.

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
