# Peer review of "Acute Effect of Riceberry Waffle Intake on Postprandial Glycemic Response in Healthy Subjects"

_foods, 2021, doi:10.3390/foods10122937_

Round 1
Reviewer 1 Report
The authors prepared a gluten free version of riceberry waffle and assessed it's glycemic response in comparison with standard formulation. This is a useful approach for expanding the availability of gluten free products. The manuscript is well written, but could be improved. The authors added artificial sweeteners and low GI sugars, but ended up with a product with a very high GI; as such it is not clear what health benefit this new product can provide.
A major problem with this report is in the method for measuring glycemic index, which was derived from testing the experimental formulation and standard glucose solution on one occasion. At minimum, the standard used for GI testing should be repeated once, with triplicate measurements recommended.
In lines 331-334; the author's comment about the large standard deviation is unfounded. In fact, the large SD is likely due to the use of the minimum number of participants (n=12 is recommended), the absence of replicate testing of the glucose standard (as recommended by Bourns et al) and the use of a glucometer to measure blood glucose.
The analysis of glycemic response and assessment of differences should be done through the use of repeat analysis of variance.
Figure 3: Y-axis has no units - this should be corrected; In the tex (lines 211-212) units for iAUC must be provided.
Line 214: please provide the SEM for the GI values
Author Response
Dear Editor and Reviewer,
Here is my revised version of manuscript and the point by point response.
Please see the attached file below. If you still have any comments or suggestion, please don't hesitate to let me know. I will do my best to edit the manuscript to be better.
Best regards,
Patthamawadee Tongkaew

Reviewer 2 Report
Please elaborate the sampling technique where the authors arrived at 10 participants. What parameters and assumptions did the authors took?
Did the participants receive any incentives? If yes - please add.
It’s still unclear to me how the blood sugar experiment is carried out. Did the 10 participants have to attend all three sessions?
It’s interesting how the authors also decided for a calorie reduced waffle - why’s that the case? Add some justification in the introduction.
Was the sample order of presentation counter balanced? Samples also 3 digit random coded?
Re sugar blood test - was there any significant differences between the samples?
Also a good practice is to include F values in the test of significance for example on table 1.
Author Response

(The authors gave the same response as above.)
